EMBO
Molecular Medicine

# Intra-arterial transplantation of HLA-matched donor mesoangioblasts in Duchenne muscular dystrophy

Giulio Cossu[1,*], Stefano C Previtali[2,3,**], Sara Napolitano[4,5], Maria Pia Cicalese[4,5], Francesco Saverio Tedesco[6], Francesca Nicastro[7,8], Maddalena Noviello[9], Urmas Roostalu[1], Maria Grazia Natali Sora[3], Marina Scarlato[3], Maurizio De Pellegrin[10], Claudia Godi[8,11], Serena Giuliani[5], Francesca Ciotti[5], Rossana Tonlorenzi[2], Isabella Lorenzetti[2], Cristina Rivellini[2], Sara Benedetti[6], Roberto Gatti[7], Sarah Marktel[5], Benedetta Mazzi[12], Andrea Tettamanti[7], Martina Ragazzi[6], Maria Adele Imro[13], Giuseppina Marano[13], Alessandro Ambrosi[14], Rossana Fiori[15], Maria Pia Sormani[16], Chiara Bonini[9], Massimo Venturini[17], Letterio S Politi[11], Yvan Torrente[8,***] & Fabio Ciceri[4,****]

## Abstract

Intra-arterial transplantation of mesoangioblasts proved safe and partially efficacious in preclinical models of muscular dystrophy. We now report the first-in-human, exploratory, non-randomized open-label phase I–IIa clinical trial of intra-arterial HLA-matched donor cell transplantation in 5 Duchenne patients. We administered escalating doses of donor-derived mesoangioblasts in limb arteries under immunosuppressive therapy (tacrolimus). Four consecutive infusions were performed at 2-month intervals, preceded and followed by clinical, laboratory, and muscular MRI analyses. Two months after the last infusion, a muscle biopsy was performed. Safety was the primary endpoint. The study was relatively safe: One patient developed a thalamic stroke with no clinical consequences and whose correlation with mesoangioblast infusion remained unclear. MRI documented the progression of the disease in 4/5 patients. Functional measures were transiently stabilized in 2/3 ambulant patients, but no functional improvements were observed. Low level of donor DNA was detected in muscle biopsies of 4/5 patients and donor-derived dystrophin in 1. Intra-arterial transplantation of donor mesoangioblasts in human proved to be feasible and relatively safe. Future implementation of the protocol, together with a younger age of patients, will be needed to approach efficacy.

**Keywords** cell therapy; Duchenne; dystrophin; mesoangioblast; MRI
**Subject Categories** Genetics, Gene Therapy & Genetic Disease; Musculoskeletal System

1   Institute of Inflammation and Repair, University of Manchester, Manchester, UK
2   Institute of Experimental Neurology (InSpe), Division of Neuroscience, IRCCS San Raffaele Scientific Institute, Milan, Italy
3   Department of Neurology, IRCCS San Raffaele Scientific Institute, Milan, Italy
4   HSR/TIGET Pediatric Clinical Research Unit, IRCCS San Raffaele Scientific Institute, Milan, Italy
5   Hematology and BMT Unit, IRCCS San Raffaele Scientific Institute, Milan, Italy
6   Department of Cell and Developmental Biology, University College London, London, UK
7   Laboratory of Analysis and Rehabilitation of Motor Function, Division of Neurosciences, IRCCS San Raffaele Scientific Institute, Milan, Italy
8   Stem Cell Laboratory, Department of Pathophysiology and Transplantation, Università degli Studi di Milano, Fondazione IRCCS Ca' Granda Ospedale Maggiore Policlinico, Milan, Italy
9   Experimental Hematology Unit, Division of Immunology, Transplantation and Infectious Diseases, IRCCS San Raffaele Scientific Institute, Milan, Italy
10  Unit of Orthopaedics, IRCCS San Raffaele Scientific Institute, Milan, Italy
11  Neuroradiology Department and Neuroradiology Research Unit, IRCCS San Raffaele Scientific Institute, Milan, Italy
12  Immunogenetics Laboratory, Department of Immunohematology & Blood Transfusion, IRCCS San Raffaele Scientific Institute, Milan, Italy
13  MolMed S.p.A., Milan, Italy
14  University San Raffaele Vita e Salute, Milan, Italy
15  Unit of Anesthesiology, IRCCS San Raffaele Scientific Institute, Milan, Italy
16  Department of Health Sciences, University of Genoa, Genoa, Italy
17  Department of Radiology, IRCCS San Raffaele Scientific Institute, Milan, Italy
    *Corresponding author. Tel: +44 1613062526; E-mail: giulio.cossu@manchester.ac.uk
    **Corresponding author. Tel: +39 226433036; E-mail: previtali.stefano@hsr.it
    ***Corresponding author. Tel: +39 255033874; E-mail: yvan.torrente@unimi.it
    ****Corresponding author. Tel: +39 226432349; E-mail: ciceri.fabio@hsr.it

## Introduction

Duchenne muscular dystrophy (DMD) is the most common muscular dystrophy, due to mutations of the X-linked dystrophin gene (Hoffman *et al*, 1987; Mercuri & Muntoni, 2013a). It causes a progressive degeneration of skeletal and cardiac muscle, leading the patient to reduced motility, wheelchair confinement, and early death, usually due to cardiac and/or respiratory failure (Muntoni *et al*, 2003; Davies & Nowak, 2006). Drug and physical therapy have extended patients' life span but only modestly its quality (Manzur *et al*, 2008; Manzur & Muntoni, 2009).

A number of new therapies for DMD have entered clinical development, and some have progressed to phase III (Benedetti *et al*, 2013; Leung & Wagner, 2013; Mercuri & Muntoni, 2013b; Ruegg, 2013; Bushby *et al*, 2014; Leung *et al*, 2014; Seto *et al*, 2014; Voit *et al*, 2014; Witting *et al*, 2014; Buyse *et al*, 2015). All appear to be safe but are often limited to a subset of patients, and long-lasting efficacy has still to be reached.

In the past, we have characterized mesoangioblasts (MABs), a subset of pericytes, from mouse, dog, and human skeletal muscle that can be expanded in culture and maintain the ability to differentiate into skeletal and smooth muscle (Minasi *et al*, 2002; Dellavalle *et al*, 2007, 2011). Noteworthy, MABs are able to cross the vessel wall when delivered intra-arterially and can thus be distributed to downstream tissues, provided that inflammation and consequent activation of the endothelium are present, as in disease progression of DMD. We tested a protocol of cell therapy in four murine and one canine model of muscular dystrophies, demonstrating safety and partial efficacy of this approach (Sampaolesi *et al*, 2003, 2006; Diaz-Manera *et al*, 2010; Tedesco *et al*, 2011, 2012; Domi *et al*, 2015).

Following an observational study in 28 DMD patients (Pts) aimed at defining longitudinally the disease progression (Lerario *et al*, 2012), five were enrolled to a "first-in-human" phase I/IIa trial of HLA-matched sibling donor MABs under immunosuppression (Eudract 2011-000176-33).

## Results

### Intra-arterial mesoangioblast infusions

Five patients (details in Table 1) underwent transplantation of MABs obtained from a muscle biopsy of an HLA-matched brother. Target dose of MABs was consistent with doses administered to dystrophic dogs in preclinical tests (Sampaolesi *et al*, 2006). All patients received at least four infusions (Appendix Fig S1) in upper and lower (Pt 01, Pt 02, Pt 03) or only lower limbs (Pt 05, Pt 06), under immunosuppression regimen (tacrolimus). The cells used as MP underwent a number of controls before infusion and showed the features reported in Appendix Table S1. In essence, they were still able of good proliferation, expressed the expected phenotype (Tonlorenzi *et al*, 2007), and had variable ability to differentiate into multinucleated myotubes in culture.

MAB infusions were in general well tolerated; however, one SAE, a thalamic stroke in Pt 03, was observed out of 23 infusions (see below and Appendix Table S2 for details).

In Pt 01 and Pt 03, cell dose was inferior to target. Immediately after MAB infusion, an asymptomatic cutaneous reticulum (*livedo reticularis*) appeared in the left abdominal lower quadrant (Fig 1A) in Pt 01 and disappeared spontaneously after 1 day; it was attributed to the infusion of small cell clumps, occasionally appearing in confluent cultures (Fig 1B). Also, Pt 02 showed a transient *livedo reticularis* after two MAB infusions (in left hand and left limb; Fig 1C). More details and a comparison with healthy children of the same age are reported in the legend to Appendix Table S2. To avoid the occurrence of cell clumps, we amended the protocol to allow filtration of the MP with a 70-μm cell strainer.

In Pt 03, during the first MAB infusion, the pre-infusion diagnostic angiography of the right lower limb revealed contrast inflow delay, likely due to vasospasm of the ipsilateral iliac–femoral arterial axis. The patient was thus infused on the contralateral patent artery after iliac crossing; the vasospasm resolved after injection of vasodilator. Pt 03 showed one SAE after the fourth (last) infusion. Five hours after MAB infusion, the Pt had an episode of vomiting and atrial fibrillation was revealed (but we do not know when it started since the Pt had not been monitored after the infusion), which resolved spontaneously one hour after having being detected. ECG, echocardiography, and color Doppler ultrasound of arteries at four limbs were all normal. The subsequent night, he had headache, photophobia, and vomiting, which solved with paracetamol. Neurological examination was normal, but brain MRI showed an acute thalamic stroke (Fig 1D). Intracranial arterial and venous MR angiography (MRA) and contrast-enhanced MRA of the supra-aortic arteries showed normal caliber and flow signal of the examined vessels. Transcranial Doppler ultrasound with micro-bubbles was normal. He was started on oral aspirin and no further complication occurred. Cerebral MRI 1 month later showed normal evolution of the ischemic lesion (Fig 1E). No new lesions or any clinical consequences were detected.

Due to the stroke in Pt 03, study Data Safety Monitoring Board (DSMB) recommended in Pt 05 and Pt 06 MAB infusions only in lower limbs for safety and with the intention to increase cell dose to reach target treatment in lower limbs. No SAEs were observed in these last patients (10 infusions).

### Donor cell engraftment and dystrophin expression

Muscle biopsies performed 2 months after the last infusion showed histological features of muscular dystrophy in all patients (Fig 2A and B). Fiber regeneration (identified by anti-fetal myosin) was minimal, ranging from 3 to 32% (Fig 2C), and rather low as compared to those usually observed in younger DMD patients (50–60%). The DNA chimerism analysis revealed minimal donor cell engraftment, ranging from 0.00 to 0.69% (Appendix Table S3).

Pt 01 and Pt 03 showed virtually no dystrophin expression by immunohistochemistry (Fig 3A). Pt 02 showed scattered, faint, dystrophin positivity in some muscle fibers in post-treatment biopsies. Fiber staining was discontinuous, but revealed also with anti-dys1 antibody, which recognizes a portion of deleted protein absent in revertant fibers (Fig 3B). Pt 05 and Pt 06 showed some fibers positive for dystrophin in both pre- and post-treatment samples (Fig 3C and D). We then applied semi-quantitative measurement of dystrophin expression levels comparing pre-treatment muscle of Pt 01 (sample of muscle obtained from the biopsy performed at time of

**Table 1.  Patient clinical features.**

|  | Pt 01 | Pt 02 | Pt 03 | Pt 05 | Pt 06 |
|---|---|---|---|---|---|
| Dystrophin mutations | Exon deletion 47–52 | Exon deletion 4–44 | Exon deletion 45–50 | Point mutation Exon 43 | Point mutation Exon 59 |
| Steroid (mg/kg) | Deflazacort 0.75/every other day | Deflazacort 0.7/every other day | Prednisone 0.5/daily | Deflazacort 0.9/every other day | Deflazacort 0.5/every other day |
| Age at first infusion (year) | 12.4 | 8.5 | 9.6 | 9.2 | 12.2 |
| Loss of ambulation (age) | 12 | 10.2 | 10.8 | Preserved walk ability | 12.1 |
| Weight (kg)[a] | 50 | 25 | 37 | 38 | 46 |
| Height (cm)[a] | 137 | 123 | 140 | 126 | 160 |
| Cardiac function | Normal | Normal | Normal | Normal | Normal |
| Lung function | Normal | Normal | Normal | Normal | Normal |
| Other comorbidities | None | None | None | None | None |

[a]Measured at first infusion.

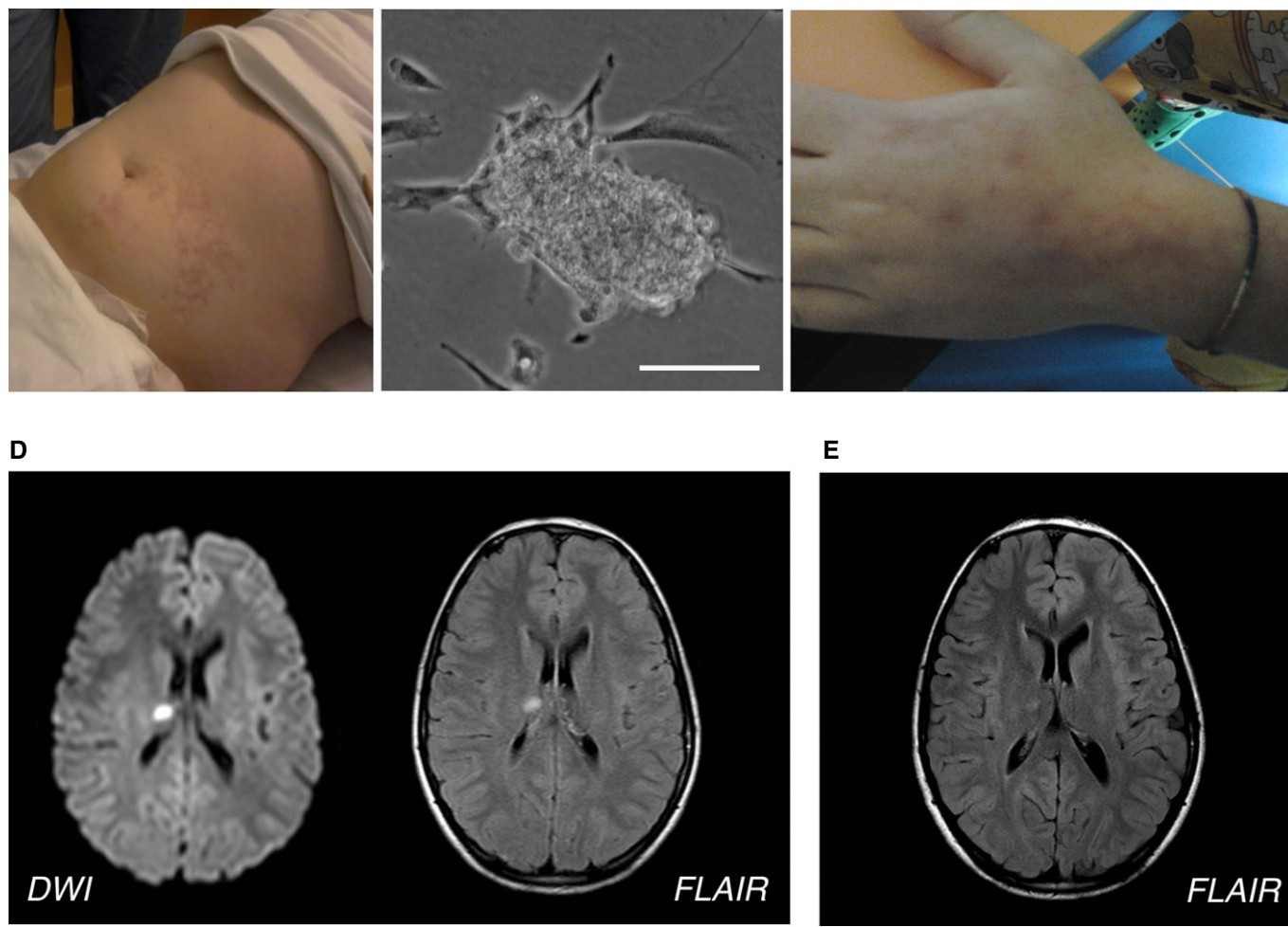

**Figure 1.  Side effects of MAB treatment of DMD patients.**

A  *Livedo reticularis* in the left abdominal lower quadrant after the first infusion in Pt 01.

B  Small clump of MABs observed in the first preparation of MP before the infusion of Pt 01. Scale bar, 30 μm.

C  *Livedo reticularis* in the left hand of Pt 02 after the first infusion.

D  Brain MRI acquired 1 day after the MAB infusion showing acute small thalamic stroke in Pt 03. Axial diffusion-weighted imaging (left) and fluid-attenuated inversion recovery (FLAIR; right) images show a focal spot of hyperintensity within the right thalamus consistent with acute stroke.

E  FLAIR MRI axial image obtained in Pt 03 1 month after the acute stroke, showing the expected evolution of the right thalamic lesion.

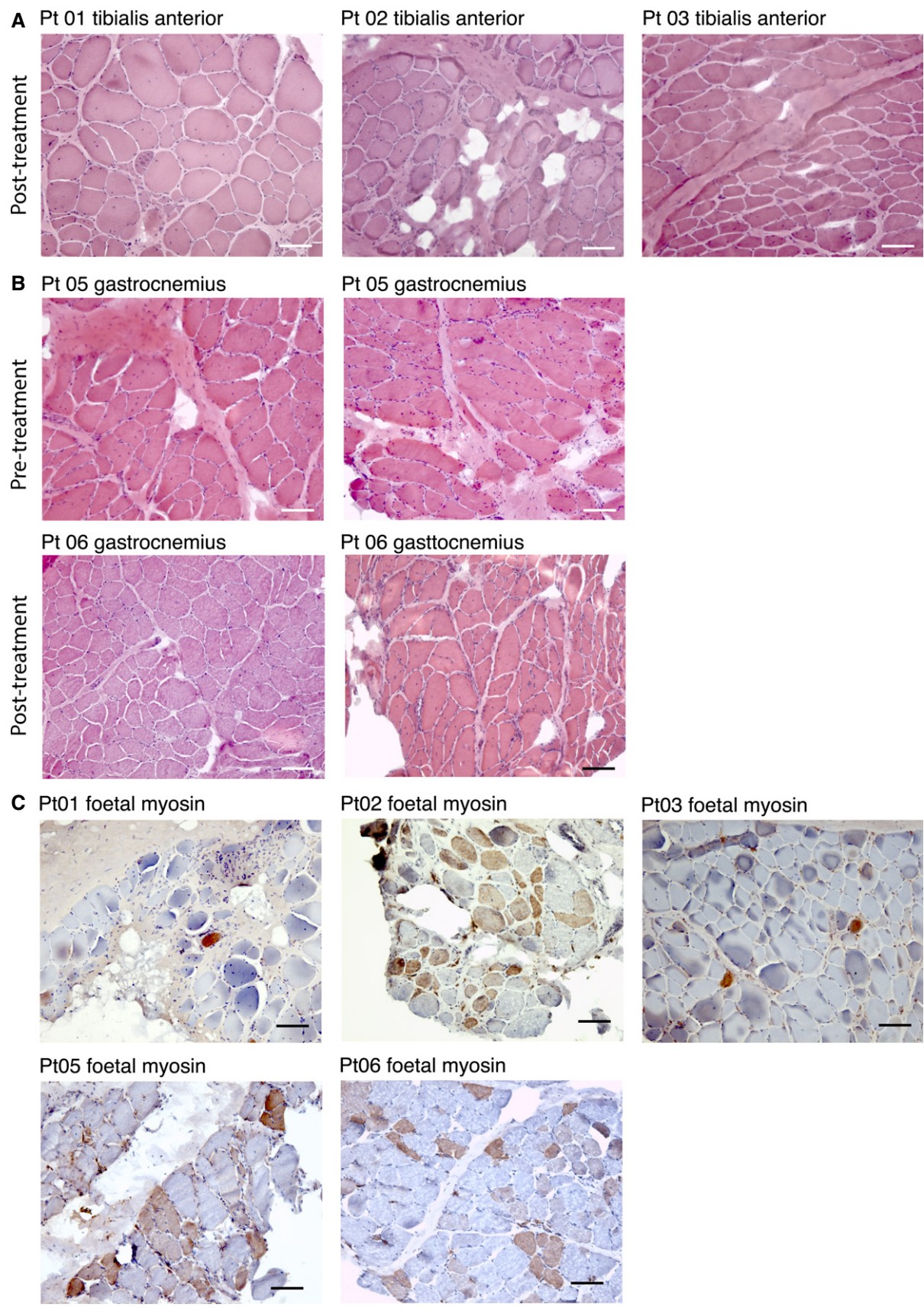

**Figure 2.**

**Figure 2.  Muscle biopsies of DMD-treated patients.**

A  Hematoxylin and eosin staining of muscle biopsies from Pt 01, Pt 02, and Pt 03 performed 2 months after the last MAB infusion. Images show diffuse increase of connective tissue, atrophic and hypertrophic fibers, degenerating fibers, and diffuse centralization of nuclei. Scale bar, 100 μm.

B  Hematoxylin and eosin staining of muscle biopsies from Pt 05 and Pt 06 performed 1 month before the first MAB infusion and 2 months after the last MAB infusion. Images show diffuse increase of connective tissue, atrophic and hypertrophic fibers, degenerating fibers, and diffuse centralization of nuclei. No evident differences were observed between biopsy performed after and before treatment. Scale bar, 100 μm.

C  Immunohistochemistry showing fetal myosin expression in muscle biopsy of DMD patients performed 2 months after the last MAB infusion. The staining was performed in tibialis anterior muscle for Pt 01, Pt 02, and Pt 03 and in gastrocnemius muscle for Pt 05 and Pt 06. A higher number of positive (brownish staining) fibers (representing regenerating fibers) were observed in Pt 02, Pt 05, and Pt 06. Scale bar, 100 μm.

diagnosis), Pt 05 and Pt 06 (muscle biopsy performed before MAB therapy) with levels in post-treatment muscle. Pt 05 showed modest post-treatment increase of dystrophin levels with anti-dys2 antibody, as mean dystrophin fluorescence intensity increased from 3 to 11% of normal control after treatment. Pt 01 and Pt 06 did not show any increase in protein expression (Appendix Fig S2). However, similar quantification with anti-dys1 antibody did not show any increase in dystrophin levels in Pt 01 and Pt 05, whereas a modest increase was observed in Pt 06 (from 11 to 22%; Appendix Fig S2).

Western blot analysis did not show any band corresponding to full-length dystrophin (427 kD) in Pt 01 (Fig 3E) and Pt 03 (data not shown). Pt 02 did not show full-length dystrophin in pre-treatment sample, whereas a faint band corresponding to full-length dystrophin was observed with anti-mandys18 and manex46e antibodies only in post-treatment samples (Fig 3F). Interestingly, mandys18 recognizes peptides from exons 17–35, deleted in Pt 02 (deletion 4–44), thus not present in revertant fibers. Pt 05 and Pt 06 (point

mutations) showed bands corresponding to dystrophin in both pre- and post-treatment samples, detected only after protein concentration with Amicon Ultra-0.5 centrifugal filter devices (Millipore) (Fig 3G and H). The amount of protein did not differ in pre- to post-treatment sample in Pt 06, whereas it was increased in post-treatment sample in Pt 05 (Fig 2G). Since Pt 02 has a deletion of 40 exons, the faint band of full molecular weight detected by Western blot may only derive from donor (MAB) cells. In the case of Pt 05 and Pt 06, the presence of a point mutation makes it impossible to distinguish between donor and revertant dystrophin by Western blot analysis. For this reason, we conducted Sanger sequencing of Pt 05 muscle cDNA, which only detected mutated cDNA (Appendix Fig S3). To increase sensitivity, we performed next-generation sequencing and a tetraprimer PCR assay (Appendix Fig S3) to detect the polymorphism. However, also these sensitive methods failed to detect donor cDNA but also ruled out skipping of the mutated allele, which was still present as the totality of the sample.

**Figure 3.  Effects of MAB treatment on dystrophin expression.**

A  Confocal immunofluorescence of muscle biopsy from Pt 01 (tibialis anterior post-treatment) stained with anti-dystrophin dys1 antibody (which recognizes protein fragment encoded by exons 26–30, green signal) and anti-laminin-2 (to delineate muscle fibers, red signal); DAPI identifies nuclei (blue signal). In the left image, only anti-dystrophin staining is shown. No dystrophin-positive fibers were observed. Scale bar, 100 μm.

B  Confocal immunofluorescence of muscle biopsy from Pt 02 (tibialis anterior post-treatment) stained with anti-dystrophin dys1 antibody (green signal) and anti-laminin-2 (red signal); DAPI identifies nuclei (blue signal). In the left image, only anti-dystrophin staining is shown. Some fiber shows mild and discontinuous dystrophin staining. Scale bar, 100 μm.

C  Immunofluorescence of muscle biopsy from Pt 05 taken before (left, gastrocnemius) and after treatment (right, gastrocnemius) stained with anti-dystrophin dys2 antibody (which recognizes exons 77–79). The number and intensity of dystrophin-positive fibers is increased in the post-treatment biopsy. Scale bar, 80 μm.

D  Immunofluorescence of muscle biopsy from Pt 06 taken before (left, gastrocnemius) and after treatment (right, gastrocnemius) stained with anti-dystrophin dys2 antibody. Few fibers show scattered dystrophin staining, without obvious differences between pre- and post-treatment samples. Scale bar, 80 μm.

E  Results of Western blot analysis involving dystrophin antibodies dys1, Mandys18, and Manex46e (recognizing respectively exons 26–30, 17–35, and 46), of total protein extracts (20 μg) obtained from post-treatment biopsy specimens of Pt 01 (tibialis anterior muscle); Ct was used as a positive dystrophin control. Below, bands corresponding to myosin heavy chain are shown as a loading control. No bands corresponding to full-length dystrophin were observed. A schematic representation of the deleted portion of the dystrophin and the region recognized by the used antibodies is depicted above the Western blot.

F  Results of Western blot analysis, involving dystrophin antibodies Mandys18 and Manex46e, of total protein extracts (10–20 μg) obtained from pre-treatment (performed at time of diagnosis) and post-treatment biopsy specimens of Pt 02 (tibialis anterior muscle); Ct was used as a positive dystrophin control, CtDmd was used as a dystrophin-negative control. Below, bands corresponding to myosin heavy chain are shown as a loading control. One faint band corresponding to full-length dystrophin is observed only in the post-treatment samples with both Mandys18 and Manex46e antibodies. A schematic representation of the deleted portion of the dystrophin and the region recognized by used antibodies is depicted above the Western blot.

G  Results of Western blot analysis, involving dystrophin antibodies Mandys106 (recognizing exon 43) and dys1, of total protein extracts (80 μg, following protein concentration by Amicon Ultra-0.5 centrifugal filter devices; Millipore) obtained from pre-treatment and post-treatment biopsy specimens of Pt 05 (gastrocnemius); Ct was used as a positive dystrophin control. Below, bands corresponding to myosin heavy chain are shown as a loading control. Bands corresponding approximately to full-length dystrophin are observed in pre- and post-treatment samples with both antibodies. However, bands in post-treatment sample appeared higher in amount. A schematic representation of the dystrophin point mutation (black vertical bar) and the region recognized by the used antibodies is depicted above the Western blot.

H  Results of Western blot analysis, involving dystrophin antibodies Mandys106 and dys1, of total protein extracts (80 μg, following protein concentration by Amicon Ultra-0.5 centrifugal filter devices, Millipore) obtained from pre-treatment and post-treatment biopsy specimens of Pt 06 (gastrocnemius); Ct was used as a positive dystrophin control. Below, bands corresponding to myosin heavy chain are shown as a loading control. Bands corresponding approximately to full-length dystrophin are observed in pre- and post-treatment samples with both antibodies. Bands in pre-treatment sample appeared higher in amount as compared to post-treatment sample. A schematic representation of the dystrophin point mutation (black vertical bar) and the region recognized by the used antibodies is depicted above the Western blot.

Source data are available online for this figure.

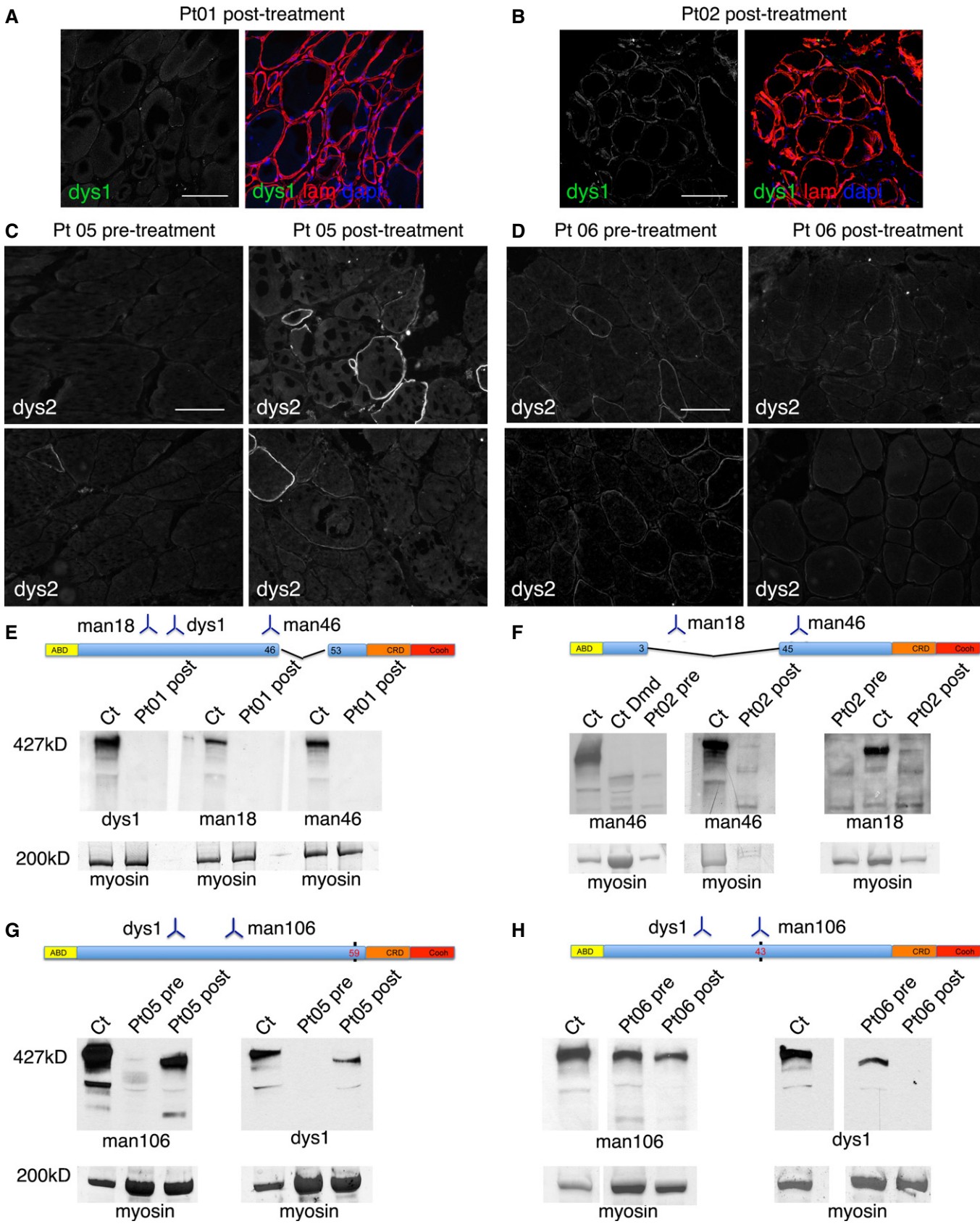

**Figure 3.**

## MRI of the lower limbs

In all MRI examinations, no signs of ischemic muscle damage, abnormal muscle edema, or ectopic/expansive lesions were detected. Signal intensity ratio (SIR) and muscle segmental volumes indexes (MVIs) obtained from the post-processing analyses of hip and lower limb muscles of transplanted patients were compared to those obtained with 85 MRI acquisitions performed on 26 untreated DMD patients who underwent a natural history study (Lerario *et al*, 2012). Four to five pre-treatment MRI examinations were available also for DMD transplanted patients (Fig 4 and Appendix Fig S4). In Pt 05, a significant post-transplant amelioration of the trend of MRI parameters was observed for SIRs of all analyzed thigh muscles (quadriceps, thigh flexor muscles, gracilis, sartorius) and for MVIs of quadriceps and semitendinosus (Fig 4); MVIs of soleus, biceps, and gracilis muscles did not reach a significant modification of the trend though a detectable improvement was observed. After treatment in Pt 06, SIRs and MVI remained relatively stable and above the 95th percentile of the disease natural history cohort (Fig 4B and Appendix Fig S4), but no significant modification of the trend was detected. No improvement was observed in Pt 01, Pt 02, and Pt 03.

## Clinical progression of the disease

All patients complied well with MAB infusions schedule and performed all planned functional and clinical tests. Respiratory and cardiac functions remained stable in all patients throughout the period of transplantation (Table 1). Pt 01 and Pt 06 were already wheelchair bound, and their motility could not be analyzed by functional tests. Pt 02 and Pt 03, although ambulant, could not perform Gowers' test already before infusions. Pt 02, who was undergoing a rapid deterioration of motor functions, appeared to stabilize, following cell infusions from April 2011 to August 2012, when he grew in height and weight, begun to limp and fall spontaneously, and within 5 months became wheelchair bound. Pt 03 progressively worsened in his quantitative and functional measures despite MAB infusions and in December 2011 lost ambulation. Finally, Pt 05 who was stable at the onset of the infusions (November 2012) remained stable until the end of 2014, when his performance worsened, possibly due to spontaneous stop of steroid (reintroduced in April 2015; Fig 5 and Appendix Fig S5). Quantitative measures with Kin-Com ergometer did not show significant changes after cell infusions in all the patients except for Pt 05. This patient showed an attenuated decrease in all the Kin-Com parameters after the infusions, as compared with his own previous slope before age 7.5 years

(Appendix Fig S6). Creatine kinase levels did not show any dose–infusion correlation in all five patients. Based on these results, immunosuppression with tacrolimus was gradually tapered and discontinued in all but Pt 05.

## Immunological studies

To evaluate the immunological effects of tacrolimus administration and multiple MAB infusions from HLA-matched siblings, immunological responses were analyzed every 2 months. Lymphocytes counts and the relative distribution of T-cell differentiation subsets remained stable and within the normal ranges throughout the study (Appendix Fig S7A–F). Importantly, in CMV- and EBV-seropositive patients, high frequencies of CMV-specific (average 103 spot-forming cells (sfc)) and EBV-specific (average 193 sfc) T-cell responses were detected by IFN-γ ELISpot throughout the study (Fig 6A), indicating that this dose of tacrolimus does not affect memory T-cell responses (Egli *et al*, 2013). Accordingly, infectious serious adverse events were not recorded.

We next investigated the development of alloreactive immune responses that could be elicited by allogeneic donor cell infusions (Noviello *et al*, 2014). MABs, MABs activated by IFN-γ (γMAB), myotubes differentiated from MABs, and PBMC harvested from each MAB donor were used as stimulators (Fig 6A). After treatment, Pt 01 showed a weak response to MAB (average 37 sfc), γMAB (average 32 sfc), and myotubes (average 18 sfc). Similarly, in Pt 06, a weak expansion of the T-cell response to γMAB was observed (average 26 sfc). In contrast, in Pt 02 and Pt 05 we did not detect alloreactive responses at any time point. Only Pt 03 showed a high basal alloreactive response before treatment: 34 sfc against MAB, 25 sfc against γMAB, and 77 sfc against myotubes. These responses rose up after treatment, and the reaction toward myotubes (158 sfc) exceeded EBV-specific response (118 sfc) at 6 m. It is noteworthy that, after *in vitro* stimulation, PBMC from Pt 03 recognized and killed donor cells, including myotubes (Appendix Fig S7H). Thus, alloreactive responses were undetectable in Pt 02 and Pt 05, while a strong response was observed in Pt 03 and week responses in Pt 01 and Pt 06.

Nearly 20% of DMD patients under steroid treatment are likely to present a specific T-cell response to dystrophin (Flanigan *et al*, 2013). Thus, we evaluated putative T-cell responses to a library of overlapping peptides covering a large portion of the dystrophin protein sequence (Fig 6B). PBMC harvested from Pt 01, Pt 02, Pt 03, and Pt 05 did not recognize dystrophin peptide pools. Pt 06 showed weak activation after challenging with pool 5, both before (16 sfc) and after treatment (14 sfc), suggesting that the dystrophin-specific

**Figure 4. Effects of MAB treatment on MRI of lower limbs.**

A  Axial T1-weighted images of the right thighs obtained in patients immediately before (upper panel) and 18 months after MAB infusion. Please note the progression of fatty degeneration and atrophy except for Pt 05 where no evident modifications can be detected.

B  Representative percentile of MRI quantitative parameters in transplanted patients compared to untreated patients. SIR (left image) and MVI (right image) of the quadriceps. Red line: Pt 01; dotted green line: Pt 02; dotted purple line: Pt 03; dotted dark blue line: Pt 05; dotted light blue line: Pt 06. Empty dots correspond to pre-treatment measures. Orange dots correspond to post-transplantation measurement. The black line corresponds to the median, while gray dotted lines to 75th, 90th, and 95th percentile. Please note the modification of the trend of Pt 05 after treatment.

C  Representative quantile spline regressions of post-transplant trend amelioration of MRI quantitative parameters in Pt 05. Upper left: SIR of the quadriceps. Lower left: SIR of thigh flexor muscles. Upper right: MVI of the quadriceps. Lower right: MVI of the semitendinosus muscles. Please note the significant modification of the patient trend (red lines) compared to median measurements obtained with 85 MRI examinations of untreated patients. Empty dots correspond to pre-treatment measures. Full dots correspond to post-transplantation measurement. The gray dotted line corresponds to the time of the first intra-arterial transplantation.

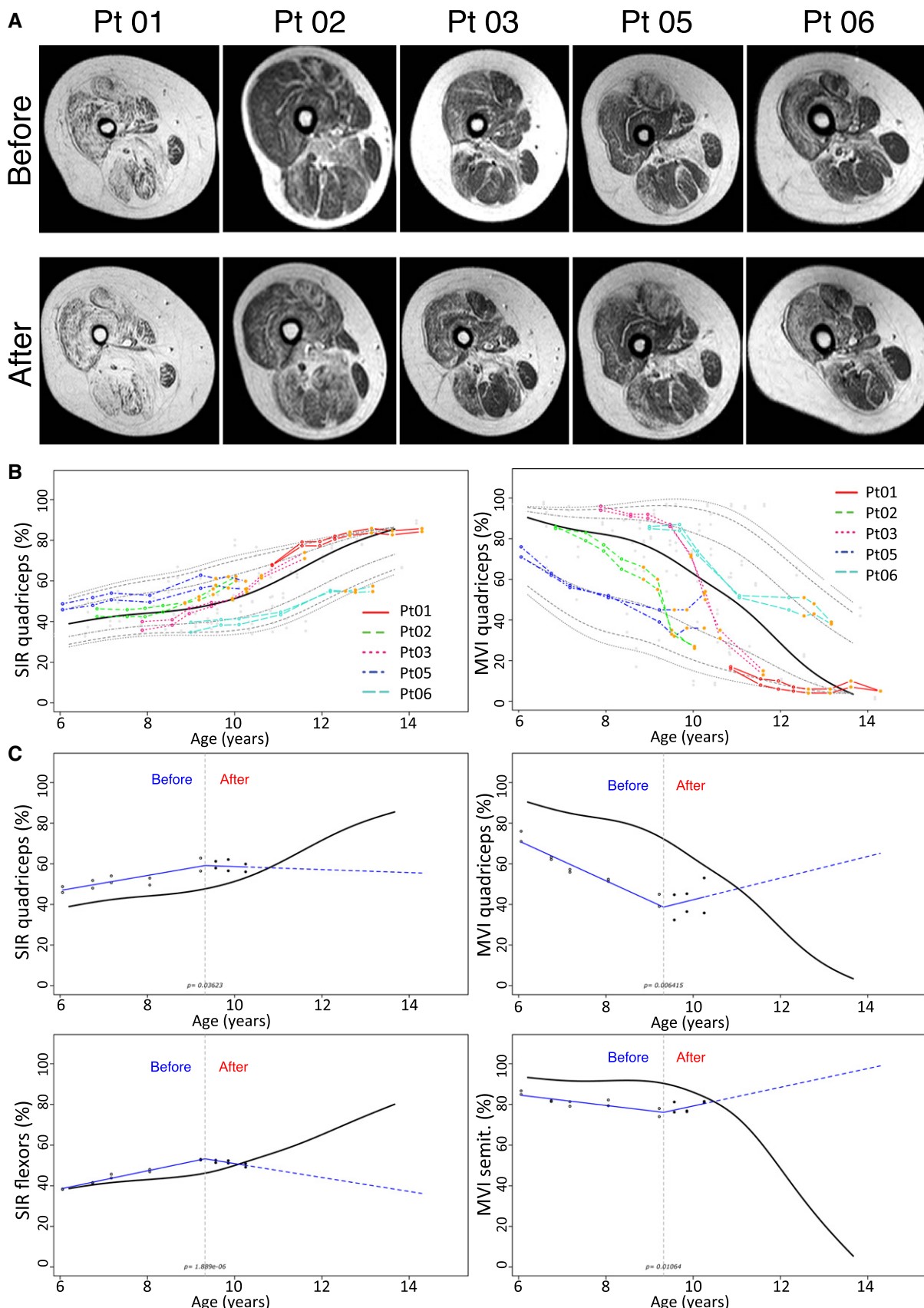

**Figure 4.**

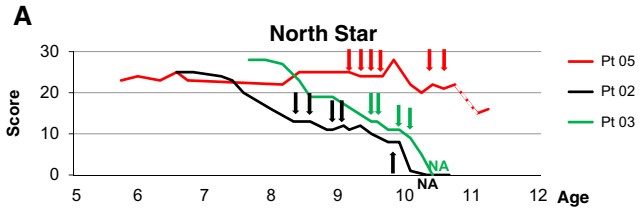

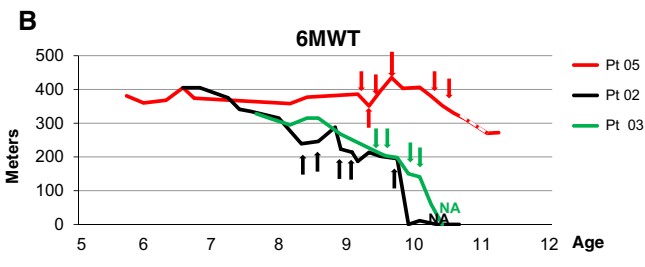

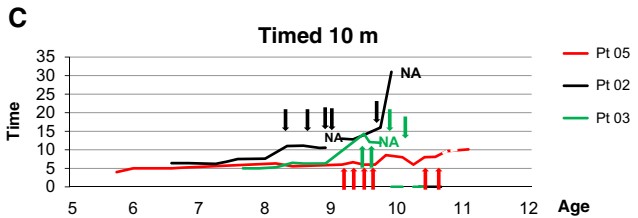

**Figure 5.  Effects of MAB treatment on outcome measures of DMD patients.**

A  North Star scale measurement (NSAA score) plotted against age of ambulant DMD patients before, throughout and after MAB clinical trial. Arrows indicate time points of MAB infusion. Dotted line indicates the time period in which Pt 05 spontaneously suspended steroids treatment without communication to parents and clinicians. Pt 02 and Pt 05 showed score stabilization throughout 8 months of MAB infusions; Pt 05 showed stabilization even in the subsequent period, whereas Pt 02 showed progressive deterioration until loss of ambulation. Pt 03 showed progressive deterioration throughout MAB infusion and lost ambulation soon after the end of the trial.

B  6-min walking test (meters) plotted against age of ambulant DMD patients before, throughout and after MAB clinical trial. Arrows indicate time points of MAB infusion. Dotted line indicates the time period in which Pt 05 spontaneously suspended steroids treatment without communication to parents and clinicians. Pt 02 and Pt 05 showed score stabilization throughout 8 months of MAB infusions; Pt 05 showed stabilization even in the subsequent period, whereas Pt 02 showed progressive deterioration until loss of ambulation. Pt 03 showed progressive deterioration throughout MAB infusion and lost ambulation soon after the end of the trial.

C  Time to run 10 m (time) plotted against age of ambulant DMD patients before, throughout and after MAB clinical trial. Arrows indicate time points of MAB infusion. Dotted line indicates the time period in which Pt 05 spontaneously suspended steroids treatment without communication to parents and clinicians. Pt 02 and Pt 05 showed time stabilization throughout 8 months of MAB infusions; Pt 05 showed stabilization even in the subsequent period, whereas Pt 02 showed progressive increase of time until loss of ambulation. Pt 03 showed progressive increase of time throughout MAB infusion and lost ambulation soon after the end of the trial.

immunity observed was not boosted by MAB infusions. Proteins extracted from a healthy donor muscle biopsy were incubated with patients' and healthy donors' sera to evaluate humoral responses to dystrophin. No circulating antibodies to dystrophin were detected (Appendix Fig S7G).

## Discussion

We report the results of a proof-of-concept phase I/IIa trial consisting multiple intra-arterial infusions of HLA-matched donor MABs in five DMD patients.

We first demonstrated the feasibility of a large-scale production under GMP conditions of MABs from a related donor of DMD patients, in fractionated doses according to a schedule of repeated therapeutic infusions. We also demonstrated that it is clinically feasible and relatively safe to inject hundreds of millions of non-hematopoietic stem cells into the peripheral arterial circulation of pediatric patients, a procedure never reported previously. This by itself has major implications for a number of different trials: Currently, there are approximately 400 clinical trials, mostly based upon intravenous administration of allogeneic or autologous mesenchymal stem cells (MSC) for a variety of disorders (Sharma *et al*, 2014). However, the vast majority of injected cells will be trapped in the lung (Schrepfer *et al*, 2007), as the first capillary filter, whereas an intra-arterial administration of these cells may direct them to the specific anatomical district favoring their engraftment and eventual function.

One SAE out of 23 infusions (4.3%), a thalamic stroke, occurred in Pt 03 after the last infusion. The cause of the thalamic stroke remains unclear; however, this event has been reported to occur spontaneously in approximately 1% of DMD patients (Hanajima & Kawai, 1996). We excluded artery dissection and embolism from patent foramen ovale. Stroke may have been caused by a cardio-embolic event during atrial fibrillation, the most common mechanism of cerebral infarction described in DMD patients (Hanajima & Kawai, 1996). Atrial fibrillation possibly followed the stress related to the procedure and to anesthesia in a DMD patient with initial signs of cardiac involvement (sinus tachycardia 92 bpm at rest) (Muntoni, 2003). Other rare possibilities include (i) migraine (Kurth *et al*, 2012), associated with 1–14% of stroke in children (Garg & DeMyer, 1995; Ebinger *et al*, 1999), (ii) minimal endothelial damage during intra-arterial catheterism, resulting in a small thrombus that, when detached, hours later caused the thalamic stroke; (iii): a small amount of MABs injected in the axillary artery due to whirlpool might have entered in the right vertebral artery, although this is unlikely due to the relevant distance between the catheter tip (in axillary artery) and the origin of the vertebral artery. Moreover, post-infusion arteriography (Appendix Fig S8) never showed stop of staining due to thrombotic events by MAB injected in high number (hundreds of millions of cells) in the four limbs. No MRI-detectable adverse events (inflammation, tumor formation, tissue infarction) occurred despite the high number of cells infused. Overall, we believe this is a relatively safe study and that the thalamic stroke was likely related to rare but already reported cardio-embolism consequent to atrial fibrillation (Hanajima & Kawai, 1996). Atrial fibrillation might be generated in predisposed patients by arterial catheterism. Minimal cardiac abnormalities should be considered as exclusion criteria together with more stringent cardiac survey in programming future studies.

The study provided important observations for MAB transplantation biology: DNA chimerism in muscles supplied by injected arteries was very low, although at expected rate in calculations of donor-to-host cells ratio; dystrophin detection was evident in the youngest treated patient. Although the study was not designed for

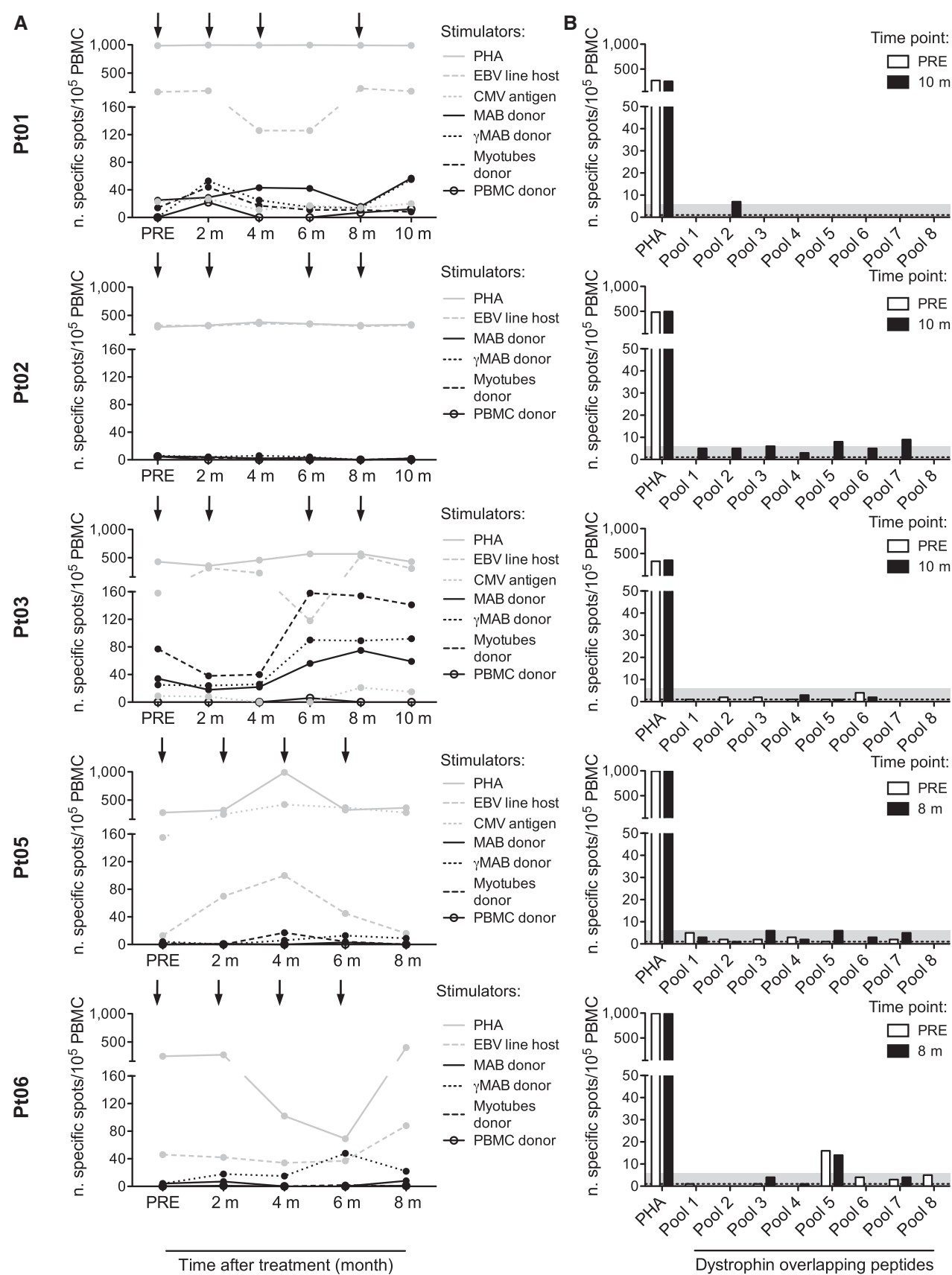

**Figure 6.**

**Figure 6.** **Effects of MAB treatment on the immune response of DMD patients.**

A   Viral and donor-specific T-cell responses were measured by IFN-γ ELISpot at different time points: before the beginning of treatment (PRE), prior to each infusion, and 2 months after the fourth infusion. Infusions are indicated by arrows. Patient's peripheral blood mononuclear cells (PBMC) were challenged with irradiated autologous lymphoblastoid cell lines (EBV line host), with cytomegalovirus glycine extract (CMV antigen) (Gehrz *et al*, 1987) and with the following irradiated cells harvested from the donor: untreated mesoangioblasts (MAB donor), MAB activated by 48-h exposure to 500 IU/ml IFN-γ (γMAB donor), myotubes differentiated from MAB (myotubes donor) and PBMC (PBMC donor). Polyclonal stimulation (phytohemagglutinin, PHA) was used as a positive control. Donor PBMC were challenged with autologous targets as negative controls. Results are expressed as number of specific cells/10$^5$ PBMC and calculated according to the following formula: number of spots produced by patient's PBMC − number of spots produced by donor's PBMC. Results for patients Pt 01, Pt 02, Pt 03, Pt 05, and Pt 06 are shown.

B   Dystrophin-specific T-cell responses were measured by IFN-γ ELISpot at two time points: before the beginning of treatment (PRE, white bar) and 2 months after the fourth infusion (black bar). PBMC were challenged with a library of 15-mer overlapping peptides covering the protein sequence (pool 1–8). Polyclonal stimulation (phytohemagglutinin, PHA) was used as a positive control. The gray area represents the threshold calculated on a cohort of healthy donors. Results for Pt 01, Pt 02, Pt 03, Pt 05, and Pt 06 are shown.

efficacy, we argue that we did not approach the threshold of MAB cell dose requested for a larger cell engraftment and consequent clinical efficacy. Older DMD patients did not change the progression of the disease upon treatment, and only in two youngest DMD patients we recorded clinical and biological indications of minimal MAB efficacy. In fact, families and physiotherapists reported less frequent falls, less rigid muscles, and a more secure ambulation in Pt 02 and Pt 05. Moreover, functional measures remained stable in Pt 02 during the infusion period (about 1 year), and Pt 05, at least 17 months after the first infusion. Pt 05 also showed other minimal but significant changes as reduced deterioration of strength at the Kin-Com ergometer and amelioration of the trend of MRI parameters, which instead did not change or deteriorated in all the other patients.

Pt 02 was the only to show unequivocal presence of donor dystrophin detected both by antibodies directed against the deleted domain and by a full molecular weight protein that he could not possibly synthesize with a deletion of 40 exons. Accordingly, in his muscles the STR chimerism analysis revealed the maximum donor cell engraftment of 0.69%, which is consistent with a billion MAB cells delivered to an approximately 8–10 kg mass of tissue, and that preclinical work indicated that about 30% of donor cells persist in the injected leg when measured after a week (Sampaolesi *et al*, 2003, 2006). The origin of increased amount of dystrophin detected in Pt 05 was clearly more complicated due to the presence of a point mutation. To rule out the possibility that it was due to revertant fibers, indistinguishable by Western blot, we performed sophisticated molecular analysis that unexpectedly revealed that all the dystrophin cDNA detected has the expected point mutation and thus comes from the patient. However, it is not derived from spontaneous exon skipping, as the exon containing the mutation was still present as the only transcript detected. We speculate that some patients (e.g., Pt 05) may have a PTC124-like endogenous mechanism that allows the production of some "full-length" dystrophin, as shown for other human genes (Schueren *et al*, 2014). Intriguingly, Pt 05 is still walking at the time of writing. However, given the paucity of donor MAB engraftment, a transient paracrine effect is possible, similar to that elicited by transplanted MSC, in a patient who likely has a slower progression of the disease on his own.

Immunological analysis revealed that immune suppression did not alter lymphocyte counts or their *in vitro* activity, and that the only cell-mediated response to donor cells was observed in Pt 03 and was detectable before MAB infusion, suggesting that this immunosuppressive regimen is adequate to prevent primary responses, but is not active on memory immunological responses.

All patients had undetectable or extremely low responses against dystrophin domains, including those absent because of large deletions.

In conclusion, our study was relatively safe (with the exception of a thalamic stroke) and provided relevant information essential for planning future cell therapy protocols in DMD patients. The lack of efficacy in treated patients as compared with preclinical work in mice and dogs may be due to several reasons: (i) the late age of patients at the onset of treatment and consequent advanced pathology of their muscles—consistently, the number of regenerating fiber was pretty low and fibrosis reduced MAB engraftment; (ii) the total cell dose that only in the last two patients matched the one administered to dogs; (iii) anti-inflammatory and immunosuppressive therapies may have reduced MAB extravasation and thus engraftment of the target tissue; (iv) substantial differences in the pathology between patients and animals; (v) the fact that humans are bipeds and need limb girdle and dorsal muscles (that we did not target) both for posture and for motility. Further trials will target younger patients, thus taking advantage of a less advanced pathology and a higher cell dose in patients before the onset of symptoms. In parallel, a cell-mediated gene correction strategy (e.g., artificial chromosomes, genome editing) that may amplify several fold the therapeutic effect and different angiographic strategies to target pelvic and dorsal muscles will be tested in animal models. We are confident that the final combination of these strategies may lead to clinical efficacy in DMD correction.

## Materials and Methods

### Patients and transplantation

The study was performed in compliance with Good Clinical Practice guidelines, the provisions of the Declaration of Helsinki and the European Directive 2001/20/EC. The San Raffaele Hospital independent ethics committee and the Istituto Superiore di Sanità approved the protocol and its subsequent amendments.

We recruited 28 patients with documented (genetic and histological) diagnosis of Duchenne muscular dystrophy, who were 6–14 years of age at the time of recruitment (March 2009), and had healthy sibling as potential donor. All the parents agreed to their participation in an observational study (DMD01) that we carried out to longitudinally follow the progression of the disease for 18 months before the onset of the interventional trial (DMD03, Eudract 2011-000176-33, March 2011). The first study (DMD01: "Outcome measures validation study for children affected by Duchenne

Muscular Dystrophy") was made to standardize outcome measures as reported (Mazzone *et al*, 2011; Lerario *et al*, 2012).

Out of these 28 DMD patients, an HLA-matched sibling donor was identified in six patients; one developed a viral myocarditis and thus failed inclusion criteria. Eligibility criteria included, beside indicated age, completion of clinical trial DMD01 and availability of potential HLA identical donor MABs, the absence of concomitant major diseases and preserved respiratory and cardiac function, absent or modest scoliosis; parental/guardian signed informed consent and patient's assent.

Five patients underwent transplantation of MABs. Target dose of MABs was consistent with doses administered to dystrophic dogs in preclinical tests (Sampaolesi *et al*, 2006). Pt 01, Pt 02, and Pt 03 were infused between March and December 2011, Pt 05 and Pt 06 between November 2012 and July 2013. Intra-arterial infusion was performed in an angiographic suite by an interventional radiologist assisted by an anesthesiologist. Before the procedure the patient received hydration with physiological solution 0.9% (1–2 ml/kg/h) via catheter-port system for 12 h. Under deep sedation (propofol 4–8 mg/kg/h), analgesia (remifentanil 0.025–0.05 mcg/kg/min) and local anesthesia (5–10 ml 1% lidocaine hydrochloride), retrograde puncture of one common femoral artery was performed and a 4F angiographic catheter was placed, with its tip either in the ipsilateral or controlateral external iliac artery and, in the last two procedures on the first three patients, also in the axillary arteries. During the procedure, all patients were monitored (ECG, HR, NBP, SaO2). A diagnostic angiography of the limb was performed before and after the infusion (Appendix Fig S8). The suspension of MABs (at a concentration of $5 \times 10^6$/ml) was infused at a rate of 1 or 2 ml/min. To avoid cell clumping, gentle and continuous agitation of the syringe was applied during infusion.

The first three patients (Pt 01, Pt 02, and Pt 03) received the first two injections in the left femoral artery at an intention dose of 3.3 and $6.6 \times 10^6$ cells/kg/body weight and the last two at $19.8 \times 10^6$ cells/kg/body weight, distributed to the other three limb arteries, so that each limb would have received a similar dose of cells in respect to its mass. Pt 05 and Pt 06 were injected exclusively in the femoral arteries, at an intention dose of 9.9, 9.9, 19.8, and $19.8 \times 10^6$ cells/kg/body weight. Pt 02 and Pt 05 received one and two additional infusions at maximal dose 1 year and 6 months after the last scheduled infusion, respectively. Only in Pt 05 and Pt 06, the target dose was reached, and the injected doses are reported in Appendix Table S4. Oral tacrolimus (0.05 mg/kg/dose twice daily, target plasma level of 5–10 μg/l), weight-adjusted steroids, acyclovir, and co-trimoxazole were administered.

Patients were weekly monitored by a local pediatrician, and at San Raffaele Hospital every 2 months during infusions and then every 3–5 months (for at least 3 years); they had a complete medical examination, including blood tests, spirometry, echocardiography, electrocardiogram, cerebral and muscle MRI, and psychological evaluation. All the severe (SAE) and minor adverse events were evaluated and classified according to National Cancer Institute (NCI) Common Terminology Criteria for adverse events (CTCAE v3.0, 2006), and eventual correlations with infusion of allo-MABs were evaluated according to the Naranjo algorithm (Naranjo *et al*, 1981).

Outcome measures to test muscle strength, functional mobility, endurance, and ability to walk included the 6-min walk test (6MWT), North Star Ambulatory Assessment (NSAA), ten-meter walk, and time to rise from floor (Gower's sign), as well as quantitative functional analysis on the Kin-Com ergometer (Mazzone *et al*, 2011; Lerario *et al*, 2012).

## Medicinal product

MolMed (http://www.molmed.com) was responsible for the production and release of MABs as medicinal product (MP). Good Manufacturing Practices (GMP) were applied to scale up the process of isolation, characterization, and expansion (Tonlorenzi *et al*, 2007). Cells were then frozen as intermediate product (IP). Cells were grown in MegaCell (Sigma–Aldrich) supplemented with five ng/ml of human recombinant basic fibroblast growth factor and 5% fetal calf serum. Features of the cells used as MP are shown in Appendix Table S1. Cells appeared morphologically homogenous and expressed the same phenotype (CD44 and CD13$^+$; CD31, CD34, and CD45$^-$) up to passage 30 at 3% O$_2$; satellite cell-derived myoblasts were minimally represented (<10% CD56$^+$) except than in one donor-derived cell population which was depleted with anti-CD56-coated magnetic beads. The MP showed myogenic potency as cells spontaneously differentiated into myotubes *in vitro* (Appendix Fig S9). Less than 6% senescent cells were detected (by beta-galactosidase expression) even at late culture passages (p29). IP cells were thawed and cultured for 1 week before the infusion, and then formulated in heparinized (5 IU/ml) normal saline solution (0.9% NaCl), constituting the fresh MP. Suitable quality control plans for IP and cells for infusion have been defined.

## Muscle biopsies and analysis of donor cell engraftment and dystrophin expression

Pt 01, Pt 02, and Pt 03 underwent muscle biopsy in the muscle *biceps brachii* and in the *tibialis anterior* 2 months after the last infusion. For Pt 01 and Pt 02, we could obtain a fragment of muscle biopsy (quadriceps) performed at time of clinical diagnosis few years before. Pt 05 and Pt 06 underwent muscle biopsy before the first infusion in the vastus lateralis, and 2 months after last infusion in vastus lateralis and gastrocnemius. All biopsies were divided in fragments for analysis of donor DNA chimerism, histology and immunohistochemistry, Western blot analysis, and RT–PCR and sequencing for dystrophin expression.

Dystrophin expression was investigated by immunohistochemistry and Western blot with anti-dys1, dys2 and dys3 (all from Novacastra, UK), Mandys18 (clone 5H9), and Manex46e (clone 8G10; from Wolfson Centre for Inherited Neuromuscular Disease, UK). For three patients, quantitative immune-fluorescence analysis was performed in double blind at ICH, University College, under the supervision of Dr. Silvia Torelli and Prof. Francesco Muntoni as described (Arechavala-Gomeza *et al*, 2010).

Regenerating fibers were identified with anti-fetal myosin antibody (Novacastra), counting at least 1000 fibers per patient (from digital images of least 6 random fields acquired with 10× objective). Light images were acquired with Olympus (BX5) microscope; conventional fluorescence with Leica DMR microscope (Leica Microsystem); and confocal images with Leica SP5 microscope (Leica Microsystem).

**Donor DNA chimerism**

Donor chimerism was evaluated by genomic detection of patient and donor-specific short tandem repeats (STR) (Kristt *et al*, 2005). Genomic DNA extraction was performed by QIAamp DNAmini kit, according to the manufacturer's recommendations (Qiagen). Chimerism analyses were performed by quantitative real-time PCR-based assay (AlleleSEQR Chimerism Assay, Abbott Molecular) using AB 7500 Real-time PCR System (Life Technologies, Thermo Fisher Scientific), and the donor percentages were obtained by the $\Delta\Delta C_t$ calculation method on triplicates for one to three informative markers.

**Sanger sequencing of 6283 C>T mutation**

Total RNA was extracted from muscle biopsy, and 1 μg was used in cDNA synthesis with random hexamer primers according to standard protocols. Subsequently, Phusion polymerase (Thermo Scientific) was used to amplify 544-bp product with oligos FDMD6283 (5′-TGCTCCTGACCTCTGTGCTA) and RDMD6283 (5′-TGACAGCTG TTTGCAGACCT). High fidelity buffer and annealing temperature 60°C was used. PCR products were purified from agarose gel using QIAquick gel extraction kit (Qiagen). Sequencing was carried out using internal primers FDMD6283seq (5′-CTACAACAAAGCTCA GGTCG) and RDMD6283seq (5′-CTCAGGAATTTGTGTCTTTCTG). Donor genomic DNA was isolated from MABs using Qiagen DNeasy Blood and Tissue Kit. Phusion polymerase with oligos FDMD6283i2 (5′-GAGCGATCCACTCTCTCAGG) and RDMD6283i3 (5′-GGAGGGT AATGCAAAGTGTAAAG) was used to amplify 841-bp product which was sequenced with internal primers FDMD6283i3 (5′-CTAC AACAAAGCTCAGGTCGGA) and RDMD6283i2 (5′-ACCTCAATGCCC CAATCTGA).

The outer primers FDMD6283 and RDMD6283 were used to amplify PCR product with Phusion DNA polymerase, which was sequenced to high depth with Illumina Miseq system at the Genomics Core Facility of the University of Manchester. The sequence reads were analyzed with the Integrative Genomics Viewer (version 2.3.30) (Robinson *et al*, 2011; Thorvaldsdottir *et al*, 2013). Nucleotide count corresponding to either 6283C or T allele is reported.

Tetraprimer ARMS PCR was carried out according to published protocols (Ye *et al*, 2001) with modifications. Primer 1 software was used to design oligos: F(C) (5′-GGGAAAAAGTTAACAAAATGTAC AAGGCCC), R(T) (5′-AACAGATCTGTCAAATCGCCCTTGGCA), F1 (5′-AACTTCTCAATGCTCCTGACCTCTGTGCT), R1 (5′-ATTCAATGT TCTGACAACAGTTTGCCGC). Similar to published protocols, we used touchdown PCR with temperature dropping 1°C per cycle from 68°C to 50°C for 18 cycles, followed by 18 cycles at 50°C. We used Platinum Taq polymerase (Life Technologies) in tetraprimer ARMS PCR.

**Safety and tolerability**

To address the risk of organ toxicity and opportunistic infections due to combined steroid and FK506 (tacrolimus) treatment, regular clinical, biochemical, drug-level, and virological (in particular CMV and EBV reactivation) monitoring was instituted, at intervals depending on patient's condition, for the entire duration of the immunosuppressive treatment and according to current standards of allogeneic stem cell transplant setting. After every MAB infusion,

patients underwent blood tests to assess muscular enzymes (CK, LDH, CK-MB, troponin T), transaminases and electrocardiography (ECG). For 24 h after every MAB infusion, a continuous monitoring of cardiac and respiratory parameters was performed.

Safety was also monitored by weekly clinical examination and bimonthly by spirometry, echocardiogram, and ECG. MRI of the lower limbs was performed before the first infusion and then every 4–6 months to assess muscular damage related to the intra-arterial injection of MABs, to register possible ischemic events and to rule out potential uncontrolled MAB expansion. All patients underwent a brain MRI examination at the end of the scheduled treatments to rule out clinically silent brain ischemic lesions.

**MRI acquisition**

MRI of the hips and lower limbs was performed immediately before, after 4, 8 and 12 months from the first MAB infusion, and then every 6 months. All MRI examinations were performed with a 3-Tesla scanner (Achieva 3T; Philips Medical Systems, Best, the Netherlands) without sedation. For the lower limbs, the scanning protocol consisted of axial T1-weighted spin echo (SE) sequences (slice thickness [THK] 4 mm; repetition time [TR] 550 ms; echo time [TE] 15 ms), axial T2-weighed turbo SE sequences (THK 4 mm; TR 7300 ms; TE 90 ms), coronal Short Tau Inversion Recovery (STIR) T2-weighted sequences (THK 5 mm; TR 7100 ms; TE 70 ms; inversion time [IT] 200 ms) and coronal 3D THRIVE T1-weighted sequences (acquired voxel size 1.25/1.25/2.50 mm, reconstructed voxel size 1.25/1.25/1.25 mm, TR 4.2 ms, TE 2.1 ms), acquired separately on the hip, thighs and calves. During the duration of the examination (26 min), a movie was projected.

All patients also underwent a brain MRI examination at the end of the treatments. For the brain MRI examination, an 8-channel phased-array dedicated coil was employed. Fluid-attenuated inversion recovery (FLAIR; THK 3 mm; TR 11,000 ms; TE 120 ms; inversion time [IT] 2800 ms), T2-weighted turbo SE (THK 4 mm; TR 3,000 ms; TE 85 ms), diffusion-weighted imaging (DWI; THK 3 mm; TR 5090 ms; TE 86 ms, b values 0–1,000 s/mm²), and inversion recovery T1 (THK 5 mm; TR 2,000 ms; TE 10 ms; IT 500 ms) sequences were acquired on the axial plane. Coronal and sagittal T2-weighted turbo SE (THK 3 mm; TR 3,000 ms; TE 85 ms) sequences were also acquired with a total scan time of 21 min, during which a movie was projected.

**MRI post-processing analysis**

*Signal intensity ratios (SIRs)*
An experienced operator manually outlined the areas of knee extensors (quadriceps), knee flexors (biceps femoris, semitendinosus and semimembranosus), sartorius, gracilis, and gluteus maximus muscles of both sides on axial T1-weighted images. The signal intensity from each region of interest (ROI) was normalized to the nearby subcutaneous fat, thus obtaining a signal intensity ratio (SIR) to measure fat infiltration. The SIR was measured on five slices covering the whole length of the femur, and then, the average was calculated.

*Lower limb segmental muscle volumes*
On axial T1-weighted images, an experienced operator manually segmented the thighs and legs volumes on a dedicated workstation

(Advantage Workstation; General Electrics, Milwaukee, USA). The anatomical landmarks for the segmentation of the volume of interest were the inter-trochanteric line and the distal femoral metaphysis for thigh, and the proximal and distal tibial metaphyses for leg. After applying a signal-intensity threshold to select muscle tissue (50–160), its volume was normalized to the previously segmented thigh and leg volumes, to obtain thigh and leg muscle volume percentage index (MVI).

*Individual muscle volumes*

The volumes of quadriceps, biceps, semitendinosus, semimembranosus, sartorius, gracilis, soleus, and gastrocnemius of both limbs were generated on axial T1-weighted images through a semi-automated system using the software Amira® (FEI Visualization Sciences Group). A signal-intensity threshold normalized to the intensity of the subcutaneous fat in each slice was then applied to measure the volume of the unaffected muscle tissue within the sheath. This volume was eventually normalized to the volume of the all muscle, obtaining percentage volumes of spared muscular tissue for each muscle.

*Statistical analyses of MRI data*

To graphically describe the pattern of natural disease progression, we considered the values of the non-treated subjects and the pre-treatment measurements of the treated ones; we fitted a nonparametric quantile regression model based on natural cubic splines as function of time. In such model, we also included the dependencies between data coming from the same subject. In particular, we considered the quantiles of order $\tau$ = 0.10, 0.25, 0.5, 0.75, and 0.90 to give a whole picture of the possible pattern distribution. The quantile regression of order $\tau$ = 0.5 gives the median tendency, which was taken as main reference in the following analysis.

Onto such plots, we plotted the values from the treated subjects, separately for leg and marked differently before–after treatment.

To further investigate the effect of treatment, for each patient we fitted the piecewise continuous linear model $y = \alpha_0 - \alpha_2 I t_0 + (\alpha_1 + \alpha_2 I)$age, with a random effect on legs in order to consider the dependencies among observations coming from the same leg, and where $t_0$ is the treatment day, $I$ is a dummy variable indicating times (age) before vs after treatment day and age is the patient's age. In this model, the parameter $\alpha_1$ gives the slope of the trend before the treatment, and $\alpha_2$ gives the change of the slope after the treatment; hence, it also gives a direct measure of the treatment effect.

**Immunological studies**

Peripheral blood mononuclear cells (PBMC) were isolated from patients by Ficoll-Hypaque gradient separation (Lymphoprep; Fresenius). Cells were cultured overnight in IMDM (GIBCO-BRL), supplemented with 10% FBS. Frequencies of T cells specific for cytomegalovirus (CMV), Epstein–Barr virus (EBV), donor PBMC, donor MABs, and donor MAB-derived myotubes were assessed by IFN-γ ELISpot and analyzed (ElyAnalyze V4.2 Reader) as described (Sacre *et al*, 2005). To further characterize alloreactive responses in Pt 03, 100-Gy-irradiated MABs obtained from the donor were used to stimulate patients' PBMC in mixed lymphocyte reactions in the presence of 60 IU/ml of rhIL2. Effectors were tested in $^{51}$Cr-release assay for the ability to kill several donor cells. To mimic

**The paper explained**

**Problem**

Muscular dystrophies are relatively common diseases that affect skeletal and often cardiac muscle. They vary in severity and age of onset but all, to variable extent, progressively compromise the ability to walk, and to move the arms, and in the most severe cases even autonomous breathing and cardiac function. As of today, there is still no effective treatment, although many new therapies are being tested in patients, some with encouraging results.

**Results**

After many years of preclinical work in several animal models, we conducted a "first-in-human" clinical trial in five patients affected by Duchenne muscular dystrophy. The trial was based upon infusion in the limb arteries of escalating doses of donor stem cells from an immunologically related brother. Clinical, instrumental, and molecular analyses were conducted before, during, and after the completion of the study. The study was relatively safe: One patient developed a thalamic stroke with no clinical consequences. However, the efficacy was minimal, possibly due to the very low number of donor cells that engrafted the patients' muscles, already compromised by the advanced stages of the disease.

**Impact**

This clinical study marks a milestone in the field of cell therapy even if it was not efficacious. For the first time, it was possible to inject hundreds of millions of non-hematopoietic stem cells in the arteries of patients without adverse events related to the procedure itself. We are now working on implementing the transplantation procedure, the efficiency of genetic correction in stem cells from the same patient aiming at applying the new protocol to younger patients, in order to approach efficacy. Importantly, the current protocol could be applied to and possibly increase the efficacy of the very many ongoing clinical trials using mesenchymal stem cells that are currently injected in the venae and are consequently trapped in the lungs.

inflammatory conditions, a fraction of MABs was exposed to IFN-γ (500 IU/ml; PeproTech) for 48 h before $^{51}$Cr labeling (Noviello *et al*, 2014). The frequencies of dystrophin-specific T cells were assessed by a 36-h IFN-γ ELISpot using as target, a library of overlapping 15-mer peptides covering the entire portion of dystrophin not overlapping with utrophin. The peptides (JPT) were divided into 8 pools, 40 peptides each, and used at a final concentration of 5 mcg/ml. PBMC harvested from 9 healthy subjects were tested as negative controls.

Absolute counts of circulating lymphocytes were quantified by flow cytometry (Beckman Coulter FC500) on the CD45$^{bright}$/SSC$^{low}$ population with fluorochrome-conjugated monoclonal antibodies to CD3, CD4, CD8, CD16, CD56, and CD19 and TruCount beads (Beckman Coulter), according to the ISCT immunological gating protocol. Data were analyzed with FCS Express (De Novo Software). The maturation phenotype of circulating T cells was further characterized by flow cytometry based on CD45RA and CD62L expression and defined as: naïve T cells (T$_{Na}$, CD45RA$^+$/CD62L$^+$), central memory T cell (T$_{CM}$, CD45RA$^-$/CD62L$^+$), effector memory T cell (CD45RA$^-$/CD62L$^-$), and effector memory T-cell expressing CD45RA (T$_{EM}$RA, CD45RA$^+$/CD62L$^-$). Humoral responses to dystrophin were evaluated by Western Blot, using proteins extracted from human skeletal muscle. The membrane was incubated with patient and donor sera at different dilutions and next with a

peroxidase-conjugated rabbit anti-human IgG (Jackson Immuno Research). The monoclonal antibody dys1 was used as positive control. The same dilutions were tested for recognition on influenza virus proteins (Vaxigrip; Sanofi Pasteur), using a monoclonal antibody to human Influenza A (H1N1, H2N2) (Takara) as control.

**Expanded view** for this article is available online.

## Acknowledgements

We thank all the patients and their family for dedication and cooperation to the DMD01 and DMD03 study; all the participating staff members of the Ospedale San Raffaele; G. Comi, N. Bresolin, A. Falini, A. Del Maschio, C. Bordignon, M.G. Roncarolo, A. Aiuti, M. Corbo, K. Gorni, M. Sessa, and M. Bregni, for their clinical and critical support; E. Mercuri, T. Mongini, R. Griggs, R. Korinthenberg, A. Madrigal, and T. Rando for their critical discussion as members of the Scientific Advisory Board; M. Moggio, A. D'Amico, and E. Bertini for offering a sample of muscle biopsy performed at the time of diagnosis of Pt 01 and Pt 02; S. Torelli and F. Muntoni for help in quantitative assessment of dystrophin expression in DMD biopsy samples; L. Callegaro, S. Trinca, M. Bonopane, and the staff members of MolMed for excellent technical assistance. Funding was provided by EC FP7 IP Optistem (Ref. 223098), Telethon Italy, Duchenne Parent Project (Italy), CureDuchenne, and the Italian Ministry of Health (progetto finalizzato (RF-2009-1547384)). G. Cossu holds a Constance Thornley professorship.

## Author contributions

GC, SCP, YT, and FC contributed to the design of the study, analysis and interpretation of the data, and writing of the manuscript. SCP, YT, MGNS, and MS recruited and analyzed patients. SCP, MDP, IL, and CR performed muscle biopsy, immunohistochemistry, and Western blot on muscle samples. SN, MPC, FC, SCP, MGNS, and SM performed clinical activity and data collection. RT, FST, SB, MR, MAI, and GM settled conditions for mesoangioblast studies and produced mesoangioblasts. FN, AT, and RG performed outcome measures. SG and FC involved in psychological management of patients and families. AA, LSP, and CG performed MRI study and interpretation of data and statistics. RF, MV, FST, RT, and GC involved in infusion of mesoangioblasts and related clinical management. CB and MN performed immunological studies. BM performed DNA chimerism studies. UR performed next-generation sequencing and a tetraprimer PCR assay studies to detect DNA polymorphism. MPS performed statistical analyses. All the authors contributed to draft and revise the manuscript and have seen and approved the final version.

## Conflict of interest

FST received a grant from Takeda New Frontier Science Programme and fee for consulting by Takeda Pharmaceuticals. MAI and GM are employed full time at MolMed. The remaining authors declare that they have no conflict of interest.

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
