## [Review Process File · EMBO Molecular Medicine]

Intra-arterial transplantation of HLA-matched donor mesoangioblasts in Duchenne Muscular Dystrophy

Giulio Cossu, Stefano C. Previtali, Sara Napolitano, Maria Pia Cicalese, Francesco Saverio Tedesco, Francesca Nicastro, Maddalena Noviello, Urmas Roostalu, Maria Grazia Natali Sora, Marina Scarlato, Maurizio De Pellegrin, Claudia Godi, Serena Giuliani, Francesca Ciotti, Rossana Tonlorenzi, Isabella Lorenzetti, Cristina Rivellini, Sara Benedetti, Roberto Gatti, Sarah Markt, Benedetta Mazzi, Andrea Tettamanti, Martina Ragazzi, Maria Adele Imro, Giuseppina Marano, Alessandro Ambrosi, Rossana Fiori, Maria Pia Sormani, Chiara Bonini, Massimo Venturini, Letterio S. Politi, Yvan Torrente, Fabio Ciceri

Corresponding authors: Giulio Cossu, University of Manchester; Stefano C. Previtali, IRCCS San Raffaele Scientific Institute, Milan; Yvan Torrente, University of Milan; Fabio Ciceri, IRCCS San Raffaele Scientific Institute, Milan

Review timeline:

Submission date:	12 July 2015
Editorial Decision:	11 September 2015
Revision received:	01 October 2015
Additional editorial correspondence	06 October 2015
Accepted:	09 October 2015

Transaction Report:

Editor: Céline Carret

1st Editorial Decision

11 September 2015

Thank you for the submission of your manuscript to EMBO Molecular Medicine and your continued patience while the paper was reviewed. We have now finally received the three reports sought that you can find copied below. You will see that the reviewers are globally supportive and I am pleased to inform you that we will be able to accept your manuscript pending the following amendments:

1) please address the minor comments from referees 2 and 3. Please note that we would strongly encourage you to provide the full western blots as requested by referee 3. Please provide a letter INCLUDING the reviewer's reports and your detailed responses to their comments (as Word file).

As you know, we encourage the publication of source data, particularly for electrophoretic gels and blots, with the aim of making primary data more accessible and transparent to the reader. Would you be willing to provide a PDF file per figure that contains the original, uncropped and unprocessed scans of all or key gels used in the figure? The PDF files should be labeled with the appropriate

figure/panel number (1 file/figure), and should have molecular weight markers; further annotation may be useful but is not essential. The PDF files will be published online with the article as supplementary "Source Data" files. If you have any questions regarding this just contact us.

Please submit your revised manuscript within two weeks.

***** Reviewer's comments *****

Referee #1 (Remarks):

Cossu and co-workers report the first- in-human application of so-called mesoangioblasts, a cell type extensively characterized by the group in previous experimental studies, in elderly patients with Duchenne's muscle dystrophy. The authors very honestly report the limited efficacy in the 5 patients treated, but clearly show safety and - to a minor extent - evidence for persistence of the cells in the recipient. They also draw the necessary conclusions to enhance potential efficacy by increasing the number of applied cells and selection of younger patients with potentially more regenerative capacity. Taken together, this is an important first step of a potentially innovative regenerative therapeutic approach for a patient cohort in desperate need for treatment.

Referee #2 (Comments on Novelty/Model System):

I felt the work was very well described, and that the authors presented a balanced interpretation of the results. Overall, the procedure itself did not appear to benefit the patients. Nonetheless, a number of important points resulted. For example, this was the first in human transplantation of a large number of non-hematopoietic stem cells into humans. The overall results will be of importance to many other groups developing cell transplantation protocols. Second, the long-term effects of tacrolimus in patients was monitored and characterized. The effects on memory T-cells were important to learn. The result that at least one patient displayed humoral immunity against dystrophin before the procedure is important and is the first confirmation of such results outside of the Flanigan lab. This alone has important implications for the design of DMD human trials. While the results did not appear to have a significant impact of the disorder, the authors presented a balanced discussion and the results will be important for future studies.

Referee #2 (Remarks):

The manuscript is well written and presents a balanced summary of the clinical trials of mesoangioblast transplantation into DMD boys. While there did not appear to be a significant clinical benefit from the procedure, it is clear, as the authors noted, that this is a precedent setting trial and the results will be of interest and importance for both follow-up studies in DMD and for other trials involving intra arterial transplantation of non-hematopoietic stem cells. The most important points seem to be the overall level of safety, the thorough characterization of immune related parameters and the multiple levels by which the results were assessed. In particular, the authors noted humoral immunity in one patient before treatment, which appears to be the first report of this outside of the Flanigan lab. That immune suppression was applied safely, without compromising memory T-cell activity was also an important point for future DMD studies. Overall the result of this study should be of significant interest to both the DMD community and the stem cell community. Overall I felt the paper was very well presented. I have only one minor comment, and that is to ask why the binding site for monoclonal antibody 106 was not shown in Figure 3E as were the other antibodies. I thought this was a nice figure and adding the last antibody would

smooth interpretation.

Referee #3 (Comments on Novelty/Model System):

This is a highly relevant work describing the first phase I/IIa clinical trial for DMD with mesoangioblasts. The findings are of great interest to the muscle community for the future implementation of therapeutic approaches for DMD. I recommend the authors address the minor points raised before publication.

Referee #3 (Remarks):

The manuscript by Cossu et al describes the outcomes of the first phase I/IIa clinical trial in 5 DMD patients using intraarterial transplantation of HLA-matched donor mesoangioblasts under immunosuppressive therapy with tacrolimus. The authors have previously shown that delivery of mesoangioblasts have proven safe and partially efficacious in preclinical studies in both mice and dogs. Patients received four cell infusions in either lower limbs or both lower and upper limbs. The work reports that the treatment was well tolerated, aside from one thalamic stroke in one patient (which resolved with no permanent injury), but the correlation to the treatment is unclear. No ischemia or edema upon cell delivery was reported. Donor cell engraftment was minimal, as shown by low level of DNA chimerism and the lack of expression of full-length dystrophin in patients. Immunological assays revealed variability in the alloreactive immune response of patients to donor cells. MRI analysis and functional tests revealed minimal to no significant improvement or modification of disease progression in the treated patients. The authors argue that this could be due to the late age of DMD patients at the time of treatment, relatively low cell dosing compared to the preclinical studies, or differences in between animal models and patients.

This is an important study as it provides evidence of safety for cell delivery by intraarterial transplantation in children, which has advantages over intramuscular or intravenous routes. It also provides a critical starting point for further improving the treatment regimen for cell-based therapies for Duchenne Muscular Dystrophy. Overall, these findings are novel and highly relevant to the skeletal muscle community.

I only have the following points that would further strengthen the manuscript:

- It would be useful to have additional details regarding the donor cells, i.e. their purity, the frequency of their spontaneous myogenicity, etc.
- It would be useful to include a discussion regarding the occurrence of infections in the treated patients reported in Table S2. Is the frequency similar to untreated patients? If not, can the author speculate as to the potential causes?
- In Figure S7G, while in the text the authors state that there was no reaction of patient sera to dystrophin, Pt 5 (Pre) and Pt 6 (8m and DN) shows dark reactivity in the whole lane, which makes it difficult to interpret. The quality of the WB should be improved. In addition the whole blot, not just cut individual bands, should be shown.

Referee #1 (Remarks):

Cossu and co-workers report the first- in-human application of so-called mesoangioblasts, a cell type extensively characterized by the group in previous experimental studies, in elderly patients with Duchenne's muscle dystrophy. The authors very honestly report the limited efficacy in the 5 patients treated, but clearly show safety and - to a minor extent - evidence for persistence of the cells in the recipient. They also draw the necessary conclusions to enhance potential efficacy by increasing the number of applied cells and selection of younger patients with potentially more regenerative capacity. Taken together, this is an important first step of a potentially innovative regenerative therapeutic approach for a patient cohort in desperate need for treatment.

We thank Referee #1 for the positive evaluation.

Referee #2 (Comments on Novelty/Model System):

I felt the work was very well described, and that the authors presented a balanced interpretation of the results. Overall, the procedure itself did not appear to benefit the patients. Nonetheless, a number of important points resulted. For example, this was the first in human transplantation of a large number of non-hematopoietic stem cells into humans. The overall results will be of importance to many other groups developing cell transplantation protocols. Second, the long-term effects of tacrolimus in patients was monitored and characterized. The effects on memory T-cells were important to learn. The result that at least one patient displayed humoral immunity against dystrophin before the procedure is important and is the first confirmation of such results outside of the Flanigan lab. This alone has important implications for the design of DMD human trials. While the results did not appear to have a significant impact of the disorder, the authors presented a balanced discussion and the results will be important for future studies.

Referee #2 (Remarks):

The manuscript is well written and presents a balanced summary of the clinical trials of mesoangioblast transplantation into DMD boys. While there did not appear to be a significant clinical benefit from the procedure, it is clear, as the authors noted, that this is a precedent setting trial and the results will be of interest and importance for both follow-up studies in DMD and for other trials involving intra arterial transplantation of non-hematopoietic stem cells. The most important points seem to be the overall level of safety, the thorough characterization of immune related parameters and the multiple levels by which the results were assessed. In particular, the authors noted humoral immunity in one patient before treatment, which appears to be the first report of this outside of the Flanigan lab. That immune suppression was applied safely, without compromising memory T-cell activity was also an important point for future DMD studies. Overall the result of this study should be of significant interest to both the DMD community and the stem cell community. Overall I felt the paper was very well presented. I have only one minor comment, and that is to ask why the binding site for monoclonal antibody 106 was not shown in Figure 3E as were the other antibodies. I thought this was a nice figure and adding the last antibody would smooth interpretation.

We thank also Referee #2 for his/her positive evaluation and have now shown the binding site for antibody 106 in the revised figure.

Referee #3 (Comments on Novelty/Model System):

This is a highly relevant work describing the first phase I/IIa clinical trial for DMD with mesoangioblasts. The findings are of great interest to the muscle community for the future implementation of therapeutic approaches for DMD. I recommend the authors address the minor points raised before publication.

Referee #3 (Remarks):

The manuscript by Cossu et al describes the outcomes of the first phase I/IIa clinical trial in 5 DMD patients using intraarterial transplantation of HLA-matched donor mesoangioblasts under immunosuppressive therapy with tacrolimus. The authors have previously shown that delivery of mesoangioblasts have proven safe and partially efficacious in preclinical studies in both mice and dogs. Patients received four cell infusions in either lower limbs or both lower and upper limbs. The work reports that the treatment was well tolerated, aside from one thalamic stroke in one patient (which resolved with no permanent injury), but the correlation to the treatment is unclear. No ischemia or edema upon cell delivery was reported. Donor cell engraftment was minimal, as shown by low level of DNA chimerism and the lack of expression of full-length dystrophin in patients. Immunological assays revealed variability in the alloreactive immune response of patients to donor cells. MRI analysis and functional tests revealed minimal to no significant improvement or modification of disease progression in the treated patients. The authors argue that this could be due to the late age of DMD patients at the time of treatment, relatively low cell dosing compared to the preclinical studies, or differences in between animal models and patients.

This is an important study as it provides evidence of safety for cell delivery by intraarterial transplantation in children, which has advantages over intramuscular or intravenous routes. It also provides a critical starting point for further improving the treatment regimen for cell-based therapies for Duchenne Muscular Dystrophy. Overall, these findings are novel and highly relevant to the skeletal muscle community.

We are also grateful to Referee #3 for the positive evaluation and have addressed below the points raised.

I only have the following points that would further strengthen the manuscript:

- It would be useful to have additional details regarding the donor cells, i.e. their purity, the frequency of their spontaneous myogenicity, etc.

We have added a new Supplementary Table (now II) detailing the phenotype of the transplanted donor cells.

- It would be useful to include a discussion regarding the occurrence of infections in the treated patients reported in Table S2. Is the frequency similar to untreated patients? If not, can the author speculate as to the potential causes?

We have added a detailed discussion of the infections occurred in treated patients in comparison with what reported in the literature for DMD patients only treated with steroids.

- In Figure S7G, while in the text the authors state that there was no reaction of patient sera to dystrophin, Pt 5 (Pre) and Pt 6 (8m and DN) shows dark reactivity in the whole lane, which makes it difficult to interpret. The quality of the WB should be improved. In addition the whole blot, not just cut individual bands, should be shown.

We apologise for the relatively poor presentation of the WB analysis. The Referee is correct in pointing out that some lanes are so dark that it is difficult to detect whether a reaction had occurred or not. This was due to the same exposure time for each blot (for logistical reasons performed at distance of years among different patients). Therefore we now show the same blots, but with different exposure times, and, as requested the whole lane. In addition we show the presence of antibodies in the serum that react with influenza virus, thus showing that the absence of a specific reactivity to dystrophin was not due to a generalised immune deficiency due to the combined action of tacrolimus and steroids.

I have now seen your responses to the referees and the latest submitted article. There were only minor issues as you know, and I will not send it back for re-review. However, before to proceed with formal acceptance, a few editorial issues have to be addressed.

[...]

Please reply by e-mail as soon as possible so we can proceed with acceptance of the paper.